# Distributed and Secure Kernel-Based Quantum Machine Learning

**Arjhun Swaminathan**[1,2,*]                              *arjhun.swaminathan@uni-tuebingen.de*

**Mete Akgün**[1,2]                                          *mete.akguen@uni-tuebingen.de*

[1]*Medical Data Privacy and Privacy-preserving Machine Learning (MDPPML), University of Tübingen*
[2]*Institute for Bioinformatics and Medical Informatics (IBMI), University of Tübingen*
[*]*Corresponding Author*

**Reviewed on OpenReview:** *https://openreview.net/forum?id=3jdIOaEW3k*

## Abstract

Quantum computing promises to revolutionize machine learning, offering significant efficiency gains for tasks such as clustering and distance estimation. Additionally, it provides enhanced security through fundamental principles like the measurement postulate and the no-cloning theorem, enabling secure protocols such as quantum teleportation and quantum key distribution. While advancements in secure quantum machine learning are notable, the development of secure and distributed quantum analogs of kernel-based machine learning techniques remains underexplored.

In this work, we present a novel approach for securely computing three commonly used kernels: the polynomial, radial basis function (RBF), and Laplacian kernels, when data is distributed, using quantum feature maps. Our methodology formalizes a robust framework that leverages quantum teleportation to enable secure and distributed kernel learning. The proposed architecture is validated using IBM's Qiskit Aer Simulator on various public datasets.

## 1 Introduction

Quantum computing is set to revolutionize machine learning (ML) by leveraging its capability to encode high-dimensional data into quantum bits, or qubits. These qubits exist in a superposition of states, enabling quantum data to represent data exponentially more efficiently than classical computing—data represented using $N$ classical bits can equivalently be represented by $log_2 N$ qubits. Further, in addition to superposition, quantum entanglement enables qubits to exhibit strong correlations that persist regardless of spatial separation. These correlations are essential for achieving exponential speedups in specific quantum algorithms (Jozsa & Linden, 2003). Although practical quantum computers are still in their infancy, various quantum machine learning (QML) techniques have been proposed (Lloyd et al., 2013; 2014; Biamonte et al., 2017).

Notably, quantum computing has exhibited substantial efficiency gains in some computational tasks compared to classical computing (Schuld & Petruccione, 2018; Zhao et al., 2021): the estimation of distances and inner products between post-processed $N$-dimensional vectors is achieved in $\mathcal{O}(log(N))$ as compared to $\mathcal{O}(N)$. Similarly, clustering $N$-dimensional vectors into $M$ clusters is expedited to $\mathcal{O}(log(MN))$ using quantum data, compared to $\mathcal{O}(poly(MN))$.

Quantum computers are not only efficient at handling high-dimensional data, but are inherently secure. This stems from two fundamental principles of quantum mechanics: the measurement postulate and the no-cloning theorem (Wootters & Zurek, 1982). Quantum data collapse upon measurement and cannot be copied without destroying the original data, offering absolutely secure communication. Secure quantum computing

is well studied and includes protocols such as quantum teleportation (Bennett et al., 1993; Bouwmeester et al., 1997), quantum key distribution (Bennett & Brassard, 2014; Bennett et al., 1992), and quantum secure direct communication (Long & Liu, 2002; Deng et al., 2003; Wang et al., 2005; Sheng et al., 2022).

Additionally, quantum computers utilizing various technologies, such as trapped ions, photons, superconducting circuits, and so forth, are actively being developed, enhancing the practical implementation of these secure communication protocols.

In parallel, kernel-based ML methods have emerged as effective tools for classification and regression tasks. These methods compute similarities between data points in high-dimensional spaces, making them particularly suited for problems where data is not trivially separable. In contrast to more advanced counterparts, such as deep learning, many kernel-based ML techniques often offer greater interpretability (Morocho-Cayamcela et al., 2019; Ponte & Melko, 2017) and better accuracy when high-dimensional data is limited, which is often the case in many real-world applications (Ding et al., 2021; Montesinos-López et al., 2021). Although much of the focus in QML has been on developing quantum or hybrid (quantum-classical) deep learning and neural networks (Garg & Ramakrishnan, 2020; Kwak et al., 2023; Dang et al., 2018; Basheer et al., 2020), quantum analogs of kernel-based ML techniques are important alternatives, with landmark studies focusing on centralized data (Havlíček et al., 2019; Schuld & Killoran, 2019).

However, real-world scenarios often involve distributed data (Yu et al., 2006; Hannemann et al., 2023), in which multiple parties wish to collaboratively train a model while ensuring the privacy of their datasets. Designing QML techniques that work securely in such distributed settings remains a critical challenge, and current research on distributed kernel-based QML methods is still sparse. In particular, Sheng & Zhou (2017) introduced a distributed framework that computes the distance between two data points for classification tasks. Later, Schuld & Killoran (2019) demonstrated that this approach is a specific instance of a more general principle: encoding data into an infinite-dimensional Hilbert space as quantum data is equivalent to mapping data into a higher-dimensional feature space for kernel computation. This insight establishes a fundamental link between quantum encoding and kernel feature maps, revealing that Sheng & Zhou (2017)'s distance computation is the quantum analog of the linear kernel within a broader theoretical context.

Building on the works of Sheng & Zhou (2017) and Schuld & Killoran (2019), our research investigates this intriguing relationship between quantum feature maps and kernel functions in the context of distributed kernel computations. Specifically, we identify appropriate quantum encodings for commonly used kernels and explore their applications in distributed learning settings. Our approach encodes classical data into quantum states using Random Fourier Features (RFF) (Rahimi & Recht, 2007) and uses a robust architecture for secure and distributed kernel-based learning. This architecture leverages quantum teleportation to ensure data security and employs a semi-honest server to compute kernels and train ML models.

We used publicly available datasets and Qiskit's Aer Simulator (Wille et al., 2019) to validate our architecture. Our evaluation demonstrates that the proposed approach ensures data security and achieves performance comparable to centralized classical and quantum methods. However, it is important to acknowledge that achieving identical or superior accuracy to classical methods is not anticipated in our study. This is not only due to widely recognized challenges in quantum computing, such as quantum noise (Bharti et al., 2022), approximations in quantum state preparation, and hardware constraints, but also due to the inherent probabilistic nature of quantum states. We make the following three contributions:

1. **Introduction of Quantum Feature Maps:** We introduce quantum feature maps for the polynomial, Radial Basis Function (RBF), and Laplacian kernels and theoretically prove the correctness of these feature maps.

2. **Architecture for Secure Kernel Computation:** We formalize a secure architecture to compute linear, polynomial, RBF, and Laplacian kernels using quantum encoding in a federated manner on distributed datasets.

3. **Implementation and Validation:** We theoretically validate our architecture, and for empirical validation, we implement it for linear kernels on publicly available datasets using Qiskit's Aer Simulator. Due to the limitations of the simulator and our lack of access to a real quantum computer, we were unable to test other kernels at this stage.

## 2 Background

### 2.1 Kernel-based Machine Learning

In machine learning, one typically works with a dataset $\mathcal{X}$ consisting of data points $\{x_1, x_2, \ldots, x_N\} \in \mathcal{X}$, where the goal is to identify patterns to evaluate previously unseen data. Kernel-based techniques employ a similarity measure, called the kernel function, between two inputs to construct models that effectively capture the underlying properties of the data distribution. This kernel function is often an inner product in a feature space, typically of higher dimensionality, where nonlinear relationships between data points become more apparent. Various kernel functions are used in practice, such as linear, RBF, polynomial, and Laplacian kernels. These functions are designed to accommodate diverse data characteristics, making them suitable for various applications. Beyond their many applications, these methods have a rich theoretical foundation, which we briefly explore below.

**Definition 1.** *Kernel function (Aizerman, 1964)*
*A* kernel function $K$ *is a map* $K : \mathcal{X} \times \mathcal{X} \to \mathbb{C}$ *that satisfies* $K(x, y) = \langle \phi(x), \phi(y) \rangle$, *where* $\phi : \mathcal{X} \to \mathcal{H}$ *is a map from the input space* $\mathcal{X}$ *to a Hilbert space* $(\mathcal{H}, \langle \cdot, \cdot \rangle)$.

One refers to $\phi$ as a feature map. Since for any unitary operator $U : \mathcal{H} \to \mathcal{H}$, $\langle \phi(x), \phi(y) \rangle = \langle U\phi(x), U\phi(y) \rangle$, a kernel can be related to many different feature maps. However, kernel theory defines a unique Hilbert space associated with a kernel, called the Reproducing Kernel Hilbert Space (RKHS) as follows.

**Definition 2.** *Reproducing Kernel Hilbert Space (RKHS) (Aronszajn, 1950)*
*Let* $\mathcal{X}$ *be an input space and* $(\mathcal{R}, \langle \cdot, \cdot \rangle)$ *the Hilbert space of functions* $f : \mathcal{X} \to \mathbb{C}$. *Then* $\mathcal{R}$ *is an RKHS if there exists a function* $K : \mathcal{X} \times \mathcal{X} \to \mathbb{C}$ *such that for all* $x \in \mathcal{X}$ *and* $f \in \mathcal{R}$, *the following holds:*

$$f(x) = \langle f, K(x, \cdot) \rangle.$$

Alternatively, considering an associated feature map, $\phi : \mathcal{X} \to \mathcal{H}$, then $\mathcal{R}$ is the space of functions $f : \mathcal{X} \to \mathbb{C}$ such that for all $x \in \mathcal{X}$ and $\nu \in \mathcal{H}$,

$$f(x) = \langle \nu, \phi(x) \rangle_{\mathcal{H}}.$$

Typically, a large family of machine learning problems aims to compute a prediction function $f : \mathcal{X} \to \mathbb{C}$ that takes training or test data and predicts the corresponding label. This is often formulated as the solution to the following optimization problem:

$$\min_{f \in \mathcal{R}} \left( \sum_{j=1}^{n} L(y_j, f(x_j)) + \lambda \Omega(f) \right), \tag{1}$$

where $L$ is a loss function, $x_j$ are training data points, $y_j$ the corresponding labels, $\lambda$ a regularization parameter controlling the trade-off between the loss and the complexity of the function $f$, and $\Omega(f)$ a general regularization term that penalizes the complexity of $f$. This prediction function generally lives in an RKHS. The representer theorem (Schölkopf & Smola, 2002) then states that the solution to this optimization problem can be formulated as follows.

$$f^*(x) = \sum_{j=1}^{n} \alpha_i K(x, x_j),$$

where $K$ is the corresponding kernel in the RKHS. Hence, optimization in an infinite-dimensional space is reduced to a finite-dimensional problem of solving for $\alpha_i$ by computing the kernel values at the training data points.

In summary, kernel functions are fundamental in machine learning, as they enable the transformation of data into higher-dimensional spaces where nontrivial relationships between data can be studied. By leveraging the theoretical framework of RKHS and the representer theorem, kernels facilitate efficient computations for a large class of machine learning models, such as Support Vector Machines (SVM), Gaussian Processes, and Principal Component Analysis (PCA) (Shawe-Taylor, 2004).

## 2.2 Random Fourier Features

Kernel methods often face significant computational challenges, particularly with large datasets. To address this issue, Rahimi & Recht (2007) introduced Random Fourier Features (RFF) as an effective approach to estimate kernel functions using finite-dimensional feature maps. RFF enables efficient computation of kernel approximations by leveraging the Fourier transform properties of shift-invariant kernels. One defines RFF as below:

**Definition 3.** *Random Fourier Features (RFF) (Rahimi & Recht, 2007)*
*Given a shift-invariant kernel $k(x - y)$ that is the fourier transform of a probability distribution $\chi$, the corresponding lower dimensional feature map $z : \mathbb{R}^D \to \mathbb{R}^d$ defined by*

$$z(x) := \left( \sqrt{2}cos(w_1 x + b_1), \ldots, \sqrt{2}cos(w_d x + b_d) \right),$$

*with $w_i \sim \chi((w_1, \ldots, w_d))$ and $b_i$ are independent samples from the uniform distribution $U[0, 2\pi]$, satisfies the following inequality for all $\epsilon$:*

$$\mathbb{P}\left( |z^T(x)z(y) - k(x,y)| \geq \epsilon \right) \leq 2\exp\left( \frac{-D\epsilon^2}{4} \right).$$

*The feature map $z$ is called an RFF.*

## 2.3 Quantum Encoding

Quantum encoding techniques are crucial for translating classical data into quantum states. There are various methods to do so, such as basis encoding, angle encoding, amplitude encoding, and Hamiltonian evolution ansatz encoding, each with its own distinct advantages and disadvantages. For example, one such encoding is defined below.

**Definition 4.** *Amplitude Encoding (Schuld et al., 2015)*
*Given classical data $x = (x_1, x_2, \ldots, x_N)^T$, where $N = 2^n$, the amplitude encoding of the data is defined as the quantum state:*

$$|\psi(x)\rangle := \sum_{j=1}^{N} \frac{x_j}{\|x\|} |j\rangle, \tag{2}$$

*where $|j\rangle$ represents the computational basis states of an n-qubit system.*

The computational basis here refers to the set of basis states that span the state space of an $n$-qubit quantum system. These states are represented with $|j\rangle$ where $j \in \{0, 1, \ldots, 2^n - 1\}$, and are expressed as tensor products of individual qubit states $|0\rangle$ and $|1\rangle$. For example, the computational basis in a 2-qubit system consists of states:

$$\{|00\rangle, |01\rangle, |10\rangle, |11\rangle\}.$$

In the context of our work, we will adopt RFF to determine the quantum encodings necessary for the computation of different kernels.

# 3 Related Work

## 3.1 Quantum Feature Maps

Building on the concept of encoding classical data into quantum states, Schuld & Killoran (2019) formalized the idea of a *quantum feature map* by noting that any quantum encoding $x \mapsto |\psi(x)\rangle$ behaves like a feature map and maps to a complex Hilbert space $\mathcal{H}$, hence naturally inducing a kernel. This opens up the possibility of utilizing the rich theory of kernel methods alongside quantum computing. Of particular interest is when two input vectors $x$ and $y$ are embedded in an $N$-dimensional Hilbert space via amplitude encoding. The resulting inner product of the encodings corresponds to the linear kernel.

$$\langle\psi(x)|\psi(y)\rangle = x^T y = k(x,y).$$

Beyond discussing the linear kernel, the work introduced additional quantum feature maps that correspond to other well-known kernels. For instance, the *Copies of Quantum States* map given by

$$x = (x_1, \ldots, x_N) \mapsto |\psi(x)\rangle = \left( \sum_{j}^{N} \frac{x_j}{\|x_j\|} |j\rangle \right)^{\otimes d}.$$

is associated with the homogeneous polynomial kernel, expressed as

$$k(x, y) = (x^T y)^d,$$

under the inner product $\langle \psi(x) | \psi(y) \rangle$. Similarly, the work proposed the following feature map -

$$x = (x_1, \ldots, x_N) \mapsto \psi(x) = \frac{1}{\sqrt{N}} \sum_{j=1}^{N} \left( \cos(x_{2j-2}) |2j-2\rangle + \sin(x_{2j-1}) |2j-1\rangle \right),$$

which is associated with the cosine kernel, expressed as

$$k(x, y) = \prod_{i=1}^{N} cos(x_i - y_i),$$

under the inner product $\langle \psi(x) | \psi(y) \rangle$.

As discussed earlier, a large family of ML algorithms optimize the functional in (1) to obtain a prediction function. There are two primary approaches to this optimization in the context of QML: the implicit approach and the explicit approach. The implicit approach uses the representer theorem and computes kernels while offloading the remaining tasks to classical computing, as demonstrated by Rebentrost et al. (2014); Schuld & Killoran (2019); Schuld (2021). The explicit approach uses variational circuits to solve the optimization problem in the infinite-dimensional RKHS, as discussed by Havlíček et al. (2019); Schuld & Killoran (2019); Cerezo et al. (2021). Our work follows the implicit approach in a distributed setting where quantum states are used for kernel computation, and the modeling is offloaded to classical computing.

### 3.2 Distributed Secure Quantum Machine Learning (DSQML)

To the best of our knowledge, only one study, Sheng & Zhou (2017), has implemented kernel-based techniques using quantum computing within a distributed framework. Their work introduced the Distributed Secure Quantum Machine Learning (DSQML) algorithm, which facilitates distance computation using a polarization-based quantum system. The setup is designed to ensure security against potential eavesdropping or interference during the computation, as any disturbance by an adversary can be detected.

The DSQML framework offers two operational modes: Client-Server and Client-Server-Database. In the Client-Server model, a client with basic quantum technology aims to classify a single data point into one of two clusters, $A$ and $B$ by computing its distance from the reference vectors $v_A$ and $v_B$. The client uses amplitude encoding to quantum encode the data point and employs quantum teleportation to delegate the inner product computation to the server. This can be interpreted as a computation of the linear kernel between the data point and the reference vectors as though in a distributed setting.

In the more complex Client-Server-Database model, the client lacks significant quantum resources and can only perform single-qubit preparation and measurement. In this setup, multiple databases encode their respective data points using amplitude encoding and transfer them to the server via quantum teleportation. The server utilizes a Fredkin gate with the client-prepared ancilla qubit as a control, performs a Hadamard gate, and returns the ancilla qubit to the client. The client then measures the qubit to extract the inner product, which is computed over multiple repetitions.

Although DSQML essentially computes the linear kernel in a distributed setting using amplitude encoding, it does not explicitly acknowledge or leverage the intrinsic relationship between quantum encoding and kernel methods as proposed later by Schuld & Killoran (2019). Consequently, it overlooked the broader kernel

framework that can be leveraged for various supervised and unsupervised machine learning tasks across different types of data, including images, text, and numeric data.

In contrast, our research substantially broadens these initial concepts by facilitating the computation of encoding-induced kernels and other standard kernels such as polynomial, RBF, and Laplacian kernels for data of any dimensionality. This generalization to other widely used kernels is much stronger and is important for future study. Further, our method follows a simple Client-Server model and can easily be adapted by relabeling to a Client-Server-Database model.

## 4 Quantum Feature Maps

Although Schuld & Killoran (2019) pointed out the implicit connection between quantum encoding techniques and feature maps, they devised feature maps only for the linear kernel, the homogeneous polynomial kernel, and the cosine kernel. We extend this by defining the quantum feature maps associated with three widely used kernels in ML: the polynomial, RBF, and Laplacian kernels.

### 4.1 Polynomial Kernel

Given classical data $x = (x_1, x_2, \ldots, x_N)^T$, we define the following quantum feature map:

$$x \mapsto \psi(x) = \bigotimes_{j=1}^{\binom{N+d}{d}} \frac{\sqrt{a}\sqrt{d!}}{\sqrt{k_1!k_2!\ldots k_{N+1}!}} x_1^{k_1} \ldots x_N^{k_N} \sqrt{c}^{k_{N+1}} |j-1\rangle, \tag{3}$$

where the multi-index $k = (k_1, \ldots, k_{N+1})$ runs over all combinations such that $\sum_{l=1}^{N+1} k_l = d$, and $c = 1 - a\|x\|$ if $d \in \mathbb{N}$, or $c = -1 - a\|x\|$ if d $\in 2\mathbb{N}$.

**Theorem 1.** *The quantum feature map above is a well-defined quantum state.*

*Proof.* To be well defined, we require the map to be normalizable. Consider the map $x \mapsto \psi(x)$. Then, using the multinomial theorem (Aizerman, 1964; Boser et al., 1992), we have that

$$\|\psi(x)\| = \sum_{\sum_l k_l = d} \left( \frac{\sqrt{a}\sqrt{d!}}{\sqrt{k_1!k_2!\ldots k_{N+1}!}} \right)^2 (x_1^{k_1})^2 \ldots (x_N^{k_N})^2 c^{k_{N+1}},$$

$$= (a\|x\| + c)^d = 1.$$

This completes the proof. $\qquad\square$

**Theorem 2.** *The quantum feature map defined above yields the polynomial kernel (Schölkopf & Smola, 2002),*

$$K_{poly} = (ax^T y + c)^d,$$

*under an inner product.*

*Proof.* This follows directly from the multinomial theorem. Let $\phi(x)$ and $\phi(y)$ be two quantum feature maps of classical data $x$ and $y$, defined as above. Then,

$$\langle \phi(x) | \phi(y) \rangle = \sum_{\sum_l k_l = d} \left( \frac{\sqrt{a}\sqrt{d!}}{\sqrt{k_1!k_2!\ldots k_{N+1}!}} \right)^2 x_1^{k_1} \ldots x_N^{k_N} y_1^{k_1} \ldots y_N^{k_N} c^{k_{N+1}},$$

$$= (ax^T y + c)^d,$$

$$= K_{poly}(x, y).$$

This completes the proof. $\qquad\square$

### 4.2 RBF Kernel

Using RFF, given classical data $x = (x_1, x_2, \ldots, x_N)^T$, we define the following quantum feature map:

$$x \mapsto \psi(x) = \frac{1}{\sqrt{D}} \sum_{j=1}^{D} \left( \cos\left(w_j^T x\right) |2j-2\rangle + \sin\left(w_j^T x\right) |2j-1\rangle \right), \tag{4}$$

where $\lceil \log_2(2D) \rceil$ determines the number of qubits used and the approximation quality, and $w_i$ are independent samples from the normal distribution $\mathcal{N}(0, \sigma^{-2}I)$.

**Theorem 3.** *The quantum feature map above is a well-defined quantum state.*

*Proof.* To be well defined, we require the map to be normalizable. Consider the map $x \mapsto \psi(x)$. Then, we have that

$$\|\psi(x)\| = \frac{1}{D} \sum_{j=1}^{D} cos^2(w_j^T x) + sin^2(w_j^T x) = 1.$$

This completes the proof. $\qquad \qquad \square$

**Theorem 4.** *The quantum feature map defined above yields the RBF kernel (Broomhead & Lowe, 1988),*

$$K_{RBF}(x, y) = \exp\left( -\frac{\|x - y\|^2}{2\sigma^2} \right),$$

*under an inner product.*

*Proof.* Let $\phi(x)$ and $\phi(y)$ be two quantum feature maps of classical data $x$ and $y$, defined as above. It follows that

$$\mathbb{E}[\langle \phi(x) | \phi(y) \rangle] = \frac{1}{D} \sum_{j=1}^{D} \mathbb{E}\left[ cos(w_j^T x) cos(w_j^T y) + sin(w_j^T x) sin(w_j^T y) \right],$$

$$= \frac{1}{D} \sum_{j=1}^{D} \mathbb{E}[\cos\left(w_j^T (x - y)\right)]. \tag{5}$$

Using Euler's formula, this can be rewritten as

$$\mathbb{E}[\langle \phi(x) | \phi(y) \rangle] = \frac{1}{2D} \sum_{j=1}^{D} \left( \mathbb{E}[\exp\left(i w_j^T (x - y)\right)] + \mathbb{E}[\exp\left(-i w_j^T (x - y)\right)] \right). \tag{6}$$

Since normal distributions remain closed under linear transformations (Wackerly et al., 2008),

$$w_j^T (x - y) = \sum_{k=1}^{N} w_{jk}(x_k - y_k) \sim N\left( 0, \frac{1}{\sigma^2} \sum_{k=1}^{N} (x_k - y_k)^2 \right) \sim \mathcal{N}\left( 0, \frac{1}{\sigma^2} \|x - y\|^2 \right).$$

Hence $w_j^T (x - y)$ is a normal distribution. Then (6) can be rewritten as

$$\mathbb{E}[\langle \phi(x) | \phi(y) \rangle] = \frac{1}{2D} \sum_{j=1}^{D} \left( M_{w_j^T (x-y)}(i) + M_{w_j^T (x-y)}(-i) \right),$$

where $M_Z(t) = \mathbb{E}[\exp\left(tZ\right)]$ is the moment generating function of a random variable $Z$. Hence, since the moment generating function of a normal distribution $Z \sim \mathcal{N}(\mu, \gamma^2)$ is given by $M_Z(t) = \exp\left(t\mu + \frac{1}{2}\gamma^2 t^2\right)$,

we have

$$\mathbb{E}[\langle \phi(x)|\phi(y)\rangle] = \frac{1}{2D} \sum_{j=1}^{D} \left[ \exp\left(-\frac{1}{2\sigma^2}\|x-y\|^2\right) + \exp\left(-\frac{1}{2\sigma^2}\|x-y\|^2\right) \right],$$

$$= \exp\left(-\frac{1}{2\sigma^2}\|x-y\|^2\right) = K_{RBF}(x,y). \tag{7}$$

This completes the proof. □

### 4.3 Laplacian Kernel

Using RFF, given classical data $x = (x_1, x_2, \ldots, x_N)^T$, we define the following quantum feature map:

$$x \mapsto \psi(x) = \frac{1}{\sqrt{D}} \sum_{j=1}^{D} \left( \cos(w_j^T x + \alpha_j) |2j-2\rangle + \sin(w_j^T x + \alpha_j) |2j-1\rangle \right), \tag{8}$$

where $\lceil \log_2(2D) \rceil$ determines the number of qubits used and the approximation quality, $w_j$ are independent samples from the Cauchy distribution $\mathcal{C}(0, \alpha^{-1}I)$, and $\alpha_j$ are independent samples from the uniform distribution $\mathcal{U}(0, 2\pi)$.

**Theorem 5.** *The quantum feature map above is a well-defined quantum state.*

*Proof.* To be well defined, we require the map to be normalizable. Consider the map $x \mapsto \psi(x)$. Then, we have that

$$\|\psi(x)\| = \frac{1}{D} \sum_{j=1}^{D} cos^2(w_j^T x + \alpha_j) + sin^2(w_j^T x + \alpha_j) = 1.$$

This completes the proof. □

**Theorem 6.** *The quantum feature map defined above yields the Laplacian kernel (Smola & Kondor, 2003),*

$$K_L(x,y) = \exp\left(-\frac{\|x-y\|_1}{\alpha}\right),$$

*under an inner product.*

*Proof.* Let $\phi(x)$ and $\phi(y)$ be two quantum feature maps of classical data $x$ and $y$, defined as above. It follows like in Theorem 4 that

$$\mathbb{E}[\langle \phi(x)|\phi(y)\rangle] = \frac{1}{2D} \sum_{j=1}^{D} \left( \mathbb{E}[\exp(iw_j^T(x-y))] + \mathbb{E}[\exp(-iw_j^T(x-y))] \right). \tag{9}$$

Since Cauchy distributions remain closed under linear transformations (Nolan, 2012),

$$w_j^T(x-y) = \sum_{k=1}^{N} w_{jk}(x_k - y_k) \sim \mathcal{C}\left(0, \frac{1}{\alpha}\sum_{k=1}^{N}|x_k - y_k|\right) \sim \mathcal{C}\left(0, \frac{\|x-y\|_1}{\alpha}\right).$$

Hence, $w_j^T(x-y)$ is a Cauchy distribution. We rewrite (9) as

$$\mathbb{E}[\langle \phi(x)|\phi(y)\rangle] = \frac{1}{2D} \sum_{j=1}^{D} \left( \phi_{w_j^T(x-y)}(1) + \phi_{w_j^T(x-y)}(-1) \right),$$

where $\phi_Z(t) = \mathbb{E}[\exp{(itZ)}]$ is the characteristic function of a random variable $Z$. Hence, since the characteristic function of a Cauchy distribution $Z \sim \mathcal{C}(\mu, \gamma)$ is given by $\phi_Z(t) = \exp{(it\mu - \gamma|t|)}$, we have

$$\mathbb{E}[\langle\phi(x)|\phi(y)\rangle] = \frac{1}{2D} \sum_{j=1}^{D} \left[\exp\left(-\frac{\|x-y\|_1}{\alpha}\right) + \exp\left(-\frac{\|x-y\|_1}{\alpha}\right)\right],$$

$$= \exp\left(-\frac{\|x-y\|_1}{\alpha}\right) = K_L(x, y). \tag{10}$$

This completes the proof. $\qquad\square$

## 5 Computational Complexity

### 5.1 Classical Setting

In classical computation, kernel evaluation involves performing pairwise operations on $N$-dimensional vectors. For example:

- The polynomial kernel requires computing the inner product - $\mathcal{O}(N)$ - and raising the result to the power $d$ - ($\mathcal{O}(d)$) - leading to a total complexity of $\mathcal{O}(N + d)$.

- The RBF kernel requires computing the squared norm of the difference between two vectors - $\mathcal{O}(N)$ - and applying the exponential function - $\mathcal{O}(1)$ - leading to a total complexity of $\mathcal{O}(N)$.

- The Laplacian kernel involves the $L_1$-norm computation ($\mathcal{O}(N)$) - and applying the exponential function - $\mathcal{O}(1)$ - leading to a total complexity of $\mathcal{O}(N)$.

This indicates that classical methods face challenges when $N$ and $d$ are large.

### 5.2 Quantum Setting

The quantum approach involves two main steps: (1) state preparation and (2) computing inner products in the corresponding Hilbert space.

**State Preparation:** Preparing the quantum feature map for an $N$-dimensional vector scales with $\mathcal{O}(N)$. This step is a bottleneck in the quantum pipeline. However, using quantum feature maps offers implicit security guarantees through the no-cloning theorem. Secure classical systems require additional overhead, such as homomorphic encryption or secure multi-party computation, to achieve comparable privacy, which can be computationally more expensive (Fan & Vercauteren, 2012).

**Inner Product Computation:** Once states are prepared, computing the inner product scales logarithmically with the dimension of the computational basis, $\bar{D}$, which determines the number of qubits required. Specifically, the number of qubits used is $\lceil\log_2 \bar{D}\rceil$, and the computational complexity is therefore $\mathcal{O}(\lceil\log_2 \bar{D}\rceil)$ for one shot. The dimension $\bar{D}$ depends on the kernel being computed:

- For polynomial kernels: $\bar{D} = \binom{N+d}{d}$, as seen in (3). Using Stirling's approximation (Stirling, 1730), the complexity can be approximated as $\mathcal{O}(d\log N)$ for large $N$ and $d$. In Appendix B.1, we show that to achieve an additive approximation error of $\epsilon$, one must use $M = \mathcal{O}(1/\epsilon^2)$ number of shots, and hence the overall complexity becomes $\mathcal{O}((d\log N) \cdot M) = \mathcal{O}((d\log N)/\epsilon^2)$.

- For RBF and Laplacian kernels: $\bar{D} = 2D$, where $D$ is the number of random Fourier features used to approximate the kernel. The number of qubits required in this case scales as $\lceil\log_2(2D)\rceil$, leading to a complexity of $\mathcal{O}(\lceil\log_2(2D)\rceil)$. In Appendices B.2 and B.3, we show that to achieve an additive approximation error of $\epsilon$ with high probability, one must choose $D = \mathcal{O}(1/\epsilon^2)$ and use $M = \mathcal{O}(1/\epsilon^2)$ shots. Consequently, the overall computational complexity for the estimation of the inner product in these cases becomes $\mathcal{O}(\lceil\log_2(2D)\rceil \cdot M) = \mathcal{O}(\lceil\log_2(2/\epsilon^2)\rceil/\epsilon^2)$.

## 6 Distributed Secure Computation of Kernels

### 6.1 Architecture

Our architecture builds on foundational concepts in quantum computing, including quantum teleportation (Bennett et al., 1993) and Fredkin gates (controlled-SWAP) (Fredkin & Toffoli, 1982), to enable secure and distributed kernel computation. While prior work in the field (Sheng & Zhou, 2017) explored similar ideas within a single-qubit/single-client framework, we extend and generalize these principles to an $n$-qubit system, $k$-client system, and thus are capable of handling high-dimensional data in diverse settings.

Our architecture comprises multiple clients, a central server, and a helper entity. The clients hold sensitive data from which they want to learn privately and collaboratively. The central server is tasked with computing the kernel securely and privately. The helper prepares entangled quantum states to facilitate quantum communication. All entities in this setup are capable of performing the necessary quantum operations. The architecture is depicted in Figure 1.

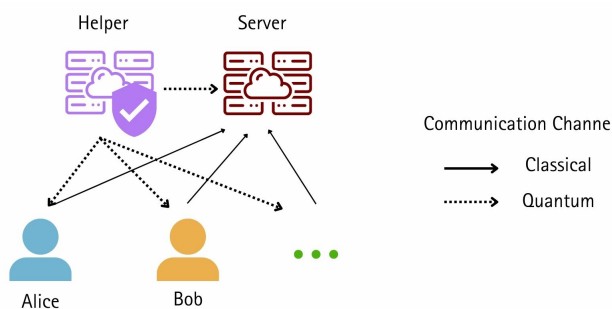

Figure 1: Visualization of our architecture consisting of a helper, a server, and multiple clients.

We employ an infrastructure in which clients are initially provided with their shared seeds securely, using established cryptographic primitives. The helper party ensures the fair distribution of seeds, adhering to standard privacy-preserving protocols.

### 6.2 Protocol Description

Without loss of generality, we describe our protocol with two participants. Our method naturally extends to any number of participants. To begin the protocol, Alice and Bob declare the size of their classical data in bits, denoted by $N$. The helper entity then computes the number of qubits, $n$, needed to encode the data for a single participant based on the chosen encoding technique. The protocol is established within a total system of $(6n + 1)$ qubits.

The protocol's circuit diagram is detailed in Figure 2 below. The correctness of the protocol is theoretically shown in Appendix A.

#### 6.2.1 Helper: Quantum State Preparation for Teleportation

The helper generates $2n$ Bell states, which are maximally entangled two-qubit states, to facilitate quantum teleportation. The helper begins by distributing the qubits between Alice, Bob, and the server as follows:

1. In the first set of $n$ Bell states, one qubit from each entangled pair represented by $|0\rangle_{HA}$ is sent to Alice, and the other represented by $|0\rangle_{SA}$ to the server.

2. In the remaining $n$ Bell states, one qubit from each entangled pair represented by $|0\rangle_{HB}$ is sent to Bob, and the other represented by $|0\rangle_{SB}$ to the server.

This enables the quantum teleportation of Alice's and Bob's encoded data to the server for secure computation.

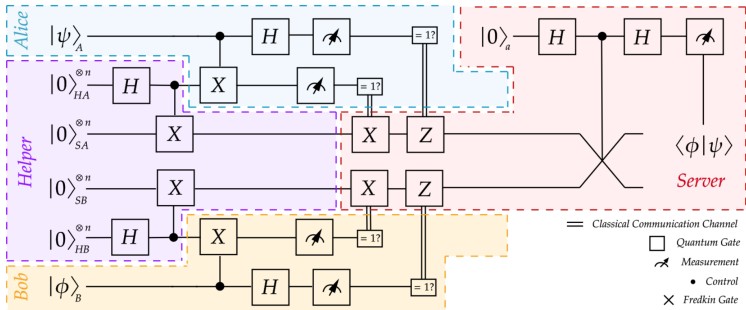

Figure 2: Quantum circuit diagram associated with our secure and distributed quantum-based kernel computation architecture.

### 6.2.2 Clients: Data Encoding and Measurement

Alice and Bob determine the encoding of their data represented by $|\psi\rangle_A$ and $|\phi\rangle_B$ respectively, with multiple encodings for every data point based on the required model accuracy. The encoding sequence is derived from the initial shared seed. Subsequently, Alice and Bob execute the following steps:

1. Apply a Controlled-X gate to the qubits they received from the helper using their original quantum data as control.

2. Apply a Hadamard gate to their original data.

3. Measure their data and the received qubits in the computational basis.

4. Communicate the results to the server through an encrypted classical communication channel.

The server applies the appropriate X and Z gates to adjust the qubits it holds.

### 6.2.3 Server: Inner Product Measurement

The server then executes a standard swap test (Barenco et al., 1997). It prepares an ancilla qubit in the zero state, applies a Hadamard gate, and uses it to conditionally swap the two sets of qubits received from Alice and Bob. After reverting the ancilla qubit with another Hadamard and measuring it, the output helps determine the required inner product (Buhrman et al., 2001). This measurement process is repeated $p$ times to enhance accuracy.

### 6.3 Security of Protocol

In our proposed adversarial model, clients, including Alice and Bob, as well as the server, are semi-honest. They adhere to the defined protocol, yet may attempt to infer additional information from the data they handle. A helper entity, deemed a semi-honest third-party, guarantees the integrity of the quantum states used in communication, similar to protocols implementing secure multi-party computation (SMPC) (Yao, 1982). The server is explicitly characterized as non-colluding with the clients, consistent with established norms in distributed and federated architectures (Swaminathan et al., 2024; Hannemann et al., 2024).

In our setup, each client processes exclusively their own data and cannot access information from other clients, effectively mitigating the risk of adversarial clients learning the data from honest input parties. The non-colluding server does not know the series of encodings applied to the original data and hence cannot reconstruct the original classical data from the quantum data it receives since it does not know how to measure it. It only learns the kernel matrix reflecting similarities between participants' data. However, since the labels are obfuscated and are irrelevant to model training, the server gains no knowledge beyond the similarity distribution pertaining to obscured labels. Further, following the same argument, we note

that in the case of the presence of colluding malicious clients and a non-colluding malicious server, the non-adversarial clients' data remain private. However, malicious participants can affect the accuracy of the model.

An adversarial third-party attempting to eavesdrop on the quantum data would face significant challenges due to the no-cloning theorem (Wootters & Zurek, 1982), which prohibits the duplication of quantum information without destroying the original information. In the event of interception, the adversarial entity would need to generate and transmit its own quantum data to the server. This can be effectively detected if clients and the server periodically exchange predetermined random quantum states, allowing the server to check for any discrepancies indicative of interference (Sheng & Zhou, 2017). Additionally, the utility of intercepted data is limited for the third-party as the encoding of data for transmission is randomized and only known to the clients through the pre-shared seed.

# 7 Experimental Evaluation

All the proof-of-concept experiments in our evaluation were carried out using classical computing resources in a High-Performance Computing (HPC) cluster. Each node within this HPC environment was equipped with an Intel XEON CPU E5-2650 v4, complemented by 256 GB of memory and 2 TB of SSD storage capacity. We used the Qiskit Aer Simulator to run the program offline due to limited access to IBM's quantum resources. As a result, we were restricted to simulating only 31 qubits in our environment. Note that we do not report any timings since the experiments are run on a simulator.

Our experiments focused on computing the linear kernel. Given the limitation of simulating only 31 qubits, which confines us to $2^7$ features, we adopted this approach and assigned $n = 7$ qubits to each party in our distributed setup. While implementing other kernels, such as encoding-induced kernels, RBF kernels, polynomial kernels, and Laplacian kernels, would require more qubits than available, our primary goal is to validate the architecture rather than exhaustively test every encoding. Since the validity of these encodings has already been theoretically established, our focus is on demonstrating that our architecture functions as expected within this framework.

In addition, we tested our methodology in a two-party configuration. This can be easily expanded, as the data can be redistributed to additional participants while maintaining consistent results. Due to the constraints on qubit simulation, a two-party configuration is employed, allocating 14 qubits for the two data providers, 14 for the helper, and 1 for the server.

## 7.1 Accuracy Analysis

We present a comparative analysis of our distributed quantum kernel learning setup against centralized quantum kernel computation and centralized classical kernel computation. Centralized quantum kernel computation only performs the swap test and does not constitute quantum teleportation. The datasets used for this analysis are widely used and publicly available. These include the Wine dataset (178 samples, 13 features) (Asuncion et al., 2007), the Parkinson's disease dataset (197 samples, 23 features) (Sakar et al., 2019), and the Framingham Heart Study dataset (4238 samples, 15 features) (Bhardwaj, 2022). Kernel-based training was performed using SVM for all datasets, and PCA was applied to the binary datasets (Parkinson's and Framingham Heart Study) to reduce dimensionality. After applying PCA, SVM was used on the transformed data to obtain accuracy metrics. All SVM training and evaluation were performed using stratified 5-fold cross-validation to ensure unbiased accuracy metrics. The accuracies of the different models on the datasets are summarized in Table 1 below.

As expected, centralized classical methods generally achieve the highest accuracy, serving as a baseline. Centralized quantum methods show competitive performance, although slightly lower than their classical counterparts, due to the inherent characteristics of quantum data and quantum simulators. Our distributed quantum architecture exhibits comparable but not the same accuracy as the centralized quantum architecture because of the complexity introduced by additional gates in the quantum circuit. All experiments were conducted with 1024 shots of the quantum circuit to ensure reliable accuracy. Here, shots refers to the number of times, $p$, a circuit is repeated.

Table 1: Comparison of accuracies across different methods and datasets.

| Dataset (Samples × Features) | Method | Accuracy | | |
|---|---|---|---|---|
| | | Centralised Classical | Centralised Quantum | Distributed Quantum |
| Wine (178 × 13) | kernel-SVM | 0.9860 ± 0.0172 | 0.8805 | 0.8874 ± 0.0259 |
| Parkinsons (197 × 23) | kernel-SVM | 0.8196 ± 0.0644 | 0.7875 | 0.7983 ± 0.0798 |
| | kernel-PCA | 0.7872 ± 0.0716 | 0.7451 | 0.7660 ± 0.0744 |
| Framingham Heart Study (4238 × 15) | kernel-SVM | 0.6788 ± 0.0108 | 0.6308 | 0.6340 ± 0.0143 |
| | kernel-PCA | 0.6788 ± 0.0095 | 0.6249 | 0.6422 ± 0.0092 |

## 7.2 Effect of Noise

Quantum computing is susceptible to various types of errors due to environmental interactions and imperfections in quantum gate implementations. Our objective is to evaluate the performance of distributed kernel-based QML under different noise conditions on Qiskit and compare it with a classical SVM. We employed three noise models:

**No Noise:** This model assumes an ideal environment without noise. It serves as a baseline for evaluating the performance of our protocol in the absence of errors.

**Noise Level 1:** This model introduces a depolarizing error with an error rate of 0.1% for single-qubit gates and two-qubit gates. The depolarizing error is a type of quantum error in which a qubit with a certain probability is replaced by a completely mixed state, losing all of its original information.

**Noise Level 2:** This model simulates a more challenging environment with a depolarizing error rate of 1%.

Our results reported in Figure 3 show that increasing noise had a negative impact on model performance.

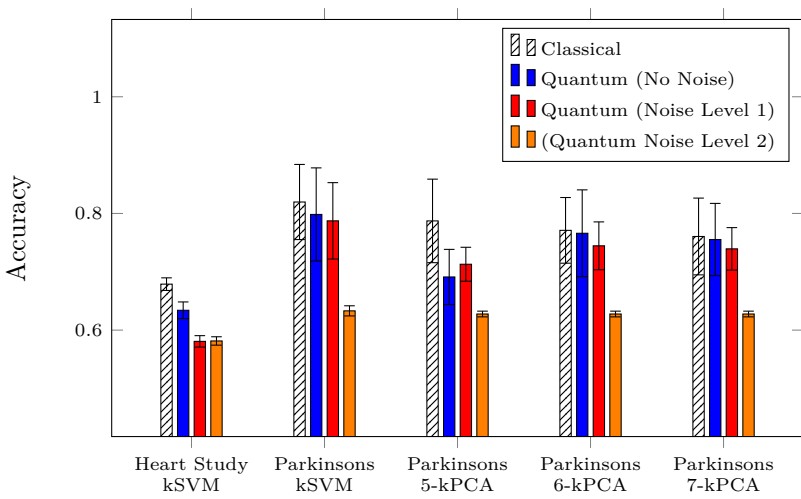

Figure 3: Comparison of accuracy scores across different noise levels. The baseline includes centralized classical kernel computation and our distributed quantum kernel computation with no noise. We incrementally introduce noise, using depolarizing error at Level 1 and Level 2, to evaluate and report the corresponding accuracy loss.

### 7.3 Effect of Shots

Here, we detail the impact of varying the number of shots used in Qiskit to repeat a quantum circuit on the performance of our proposed algorithm. We used a subset of the Digits dataset containing 100 samples (Pedregosa et al., 2011). The objective was to classify these samples into 10 labels (0-9) and to evaluate the classification accuracy using linear kernel-based SVM.

We varied the number of shots, specifically using 128, 256, 512, and 1024 shots, to observe the effect on the classification accuracy. The results, depicted in Figure 4, indicate improved performance with an increased number of shots.

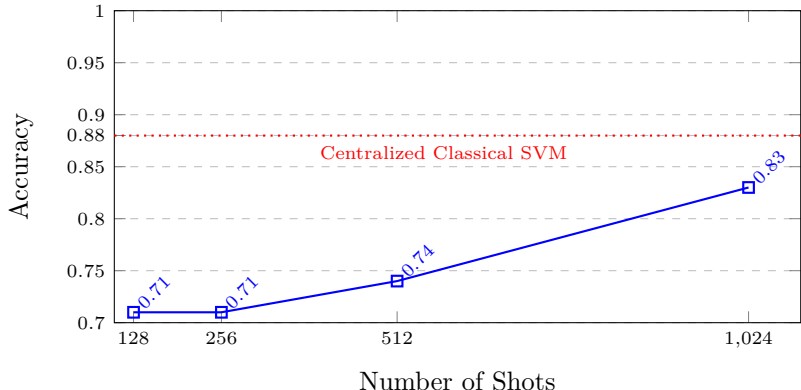

Figure 4: Accuracy of linear kernel-based SVM on a subset of the Digits dataset featuring 100 samples and 10 labels, compared against the number of shots run by the simulator.

## 8 Conclusion and Future Work

In this study, we introduced a novel kernel-based QML algorithm that operates within a distributed and secure framework. Using the implicit connection between quantum encoding and kernel computation, our method supports the calculation of encoding-induced and traditional kernels. To that end, we extended the existing body of knowledge by introducing three novel quantum feature maps designed to compute the polynomial, RBF, and Laplacian kernels, building upon previous studies that primarily focused on linear and homogeneous polynomial kernels. Furthermore, we have theoretically validated that the proposed quantum feature maps help to compute the concerned kernels.

Using a hybrid quantum-classical architecture, our approach functions in a distributed environment where a central server assists data providers in processing their data collaboratively, akin to a centralized model. This setup primarily addresses the case of a semi-honest scenario—a common consideration in studies involving distributed architectures. We have demonstrated that our proposed framework upholds security against semi-honest parties as well as external eavesdroppers.

The application of our architecture to compute the linear kernel for publicly available datasets using Qiskit's Aer Simulator validates our distributed framework, yielding accuracies comparable to those achieved in centralized classical and quantum frameworks.

In the future, our aim is to adapt our methodology to scenarios that involve actively malicious entities. Additionally, given the rich potential of kernel theory, further theoretical and practical exploration of quantum feature maps represents a promising direction for future research in the field.

### Supplementary information

Our code and results are available at the following URL: `https://github.com/mdppml/distributed-secure-kernel-based-QML`.

## Acknowledgments

This research was supported by the German Federal Ministry of Education and Research (BMBF) under project number 01ZZ2010 and partially funded through grant 01ZZ2316D (PrivateAIM). The authors acknowledge the usage of the Training Center for Machine Learning (TCML) cluster at the University of Tübingen.

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

## A    Correctness of Architecture

In this section, we provide a theoretical proof of correctness for our architecture, demonstrating that it accurately computes the kernel matrix for given input data. Without loss of generality, consider an $n$-qubit system, where Alice's encoded data is represented by $|\psi\rangle_A$ and Bob's data by $|\psi\rangle_B$. As illustrated in Figure 1, the Helper initializes the system by preparing $2n$ Bell states. The initial state consists of $|0\rangle_{HA}^{\otimes n}$, $|0\rangle_{SA}^{\otimes n}$, $|0\rangle_{SB}^{\otimes n}$, and $|0\rangle_{HB}^{\otimes n}$, where the superscript denotes the qubits in each subsystem, e.g., the $i$-th qubit of Alice's data is represented as $|\psi\rangle_A^i$.

Let $|\psi\rangle$ be written as $\alpha|0\rangle + \beta|1\rangle$ and $|\phi\rangle$ as $\delta|0\rangle + \gamma|1\rangle$ in the computational basis. Initially, the entire system is in the state:

$$|\psi\rangle_A^{\otimes n} \otimes |0\rangle_{HA}^{\otimes n} \otimes |0\rangle_{SA}^{\otimes n} \otimes |0\rangle_{SB}^{\otimes n} \otimes |0\rangle_{HB}^{\otimes n} \otimes |\phi\rangle_B^{\otimes n}.$$

For simplicity, we track only the $i$-th qubit of each $n$-qubit state:

$$|\psi\rangle_A^i \otimes |0\rangle_{HA}^i \otimes |0\rangle_{SA}^i \otimes |0\rangle_{SB}^i \otimes |0\rangle_{HB}^i \otimes |\phi\rangle_B^i.$$

After applying Hadamard gates to the Helper qubits, the system evolves to the following state:

$$|\psi\rangle_A^i \otimes \frac{1}{\sqrt{2}} \left( |0\rangle_{HA}^i + |1\rangle_{HA}^i \right) \otimes |0\rangle_{SA}^i \otimes |0\rangle_{SB}^i \otimes \frac{1}{\sqrt{2}} \left( |0\rangle_{HB}^i + |1\rangle_{HB}^i \right) \otimes |\phi\rangle_B^i.$$

Upon applying the Controlled-X gates, the system is entangled, preparing it for quantum teleportation:

$$\frac{1}{2} \left( |\psi\rangle_A^i \otimes \left( |00\rangle_{HA,SA}^i + |11\rangle_{HA,SA}^i \right) \otimes \left( |00\rangle_{SB,HB}^i + |11\rangle_{SB,HB}^i \right) \otimes |\phi\rangle_B^i \right).$$

Next, Alice and Bob perform Controlled-X gates, resulting in the state:

$$\frac{1}{2} \left( \alpha|000\rangle_{A,HA,SA}^i + \beta|110\rangle_{A,HA,SA}^i + \alpha|011\rangle_{A,HA,SA}^i + \beta|101\rangle_{A,HA,SA}^i \right)$$
$$\otimes \frac{1}{2} \left( \gamma|000\rangle_{SB,HB,B}^i + \delta|011\rangle_{SB,HB,B}^i + \gamma|110\rangle_{SB,HB,B}^i + \delta|101\rangle_{SB,HB,B}^i \right).$$

After applying Hadamard gates, the system evolves to:

$$\frac{1}{4} \left( \alpha|000\rangle_{A,HA,SA}^i + \alpha|100\rangle_{A,HA,SA}^i + \beta|010\rangle_{A,HA,SA}^i - \beta|110\rangle_{A,HA,SA}^i \right.$$
$$\left. + \alpha|011\rangle_{A,HA,SA}^i + \alpha|111\rangle_{A,HA,SA}^i + \beta|001\rangle_{A,HA,SA}^i - \beta|101\rangle_{A,HA,SA}^i \right)$$
$$\otimes \frac{1}{4} \left( \gamma|000\rangle_{SB,HB,B}^i + \gamma|001\rangle_{SB,HB,B}^i - \delta|011\rangle_{SB,HB,B}^i + \delta|010\rangle_{SB,HB,B}^i \right.$$
$$\left. + \gamma|110\rangle_{SB,HB,B}^i + \gamma|111\rangle_{SB,HB,B}^i - \delta|101\rangle_{SB,HB,B}^i + \delta|100\rangle_{SB,HB,B}^i \right).$$

Once the classical bits are communicated to the server, the server applies the appropriate X and Z gates, resulting in the system state:

$$|0\rangle_a \left(\alpha |0\rangle_{SA} + \beta |1\rangle_{SA}\right) \otimes \left(\gamma |0\rangle_{SB} + \delta |1\rangle_{SB}\right).$$

After applying a Hadamard gate, the server obtains:

$$\frac{1}{\sqrt{2}}(|0\rangle_a + |1\rangle_a)(|\psi\rangle_{SA}) \otimes (|\phi\rangle_{SB}).$$

The final step involves applying Fredkin gates, with the ancilla qubit as the control:

$$\frac{1}{\sqrt{2}} \left( |0\psi\phi\rangle_{a,SA,SB} + |1\phi\psi\rangle_{a,SA,SB} \right).$$

Upon applying a Hadamard gate to the ancilla qubit, we obtain:

$$\frac{1}{2} \left( |0\rangle_a \otimes (|\psi\phi\rangle_{SA,SB} + |\phi\psi\rangle_{SA,SB}) + |1\rangle_a \otimes (|\psi\phi\rangle_{SA,SB} - |\phi\psi\rangle_{SA,SB}) \right).$$

Measuring the ancilla qubit along the computational basis yields:

$$Pr(0)_a = \frac{1}{4} \left( \langle\psi| \langle\phi| + \langle\phi| \langle\psi| \right) \left( |\psi\rangle |\phi\rangle + |\phi\rangle |\psi\rangle \right) = \frac{1}{2} + \frac{1}{2} \|\langle\psi|\phi\rangle\|^2,$$

where, $P(0)_a$ is the probability that the anscilla qubit is in the $|0\rangle$ state. Rearranging, this then determines the inner product of $|\psi\rangle$ and $|\phi\rangle$.

$$\|\langle\psi|\phi\rangle\| = \sqrt{2Pr(0)_a - 1}, \tag{11}$$

## B   Error Analysis and Measurement Complexity

### B.1   Shot Complexity for Inner Product Estimation

In this section, we derive the relationship between the number of measurement shots, $M$, and the additive error $\epsilon$ in estimating the inner product of quantum states.

As discussed earlier in (11), the probability of obtaining the outcome $|0\rangle$ in the inner product measurement is

$$p = \frac{1 + \|\langle\psi|\phi\rangle\|^2}{2}.$$

For each shot, we define the Bernoulli random variable $X_i$ by

$$X_i = \begin{cases} 1, & \text{if the outcome is } |0\rangle, \\ 0, & \text{if the outcome is } |1\rangle. \end{cases}$$

We know $\mathbb{E}[X_i] = p$ and $\text{Var}(X_i) = p(1-p)$. After $M$ independent shots, the sample mean is $\bar{X} = \sum_{i=1}^{M} X_i/M$, with $\mathbb{E}[\bar{X}] = p$ and $\text{Var}(\bar{X}) = (p(1-p)/M)$. Labeling $(1 + \|\langle\psi|\phi\rangle\|)^2/2 = l$, an estimator for $l$ is $\hat{l} = 2\bar{X} - 1$. Its variance is $\text{Var}(\hat{l}) = 4 \text{Var}(\bar{X}) = (4p(1-p))/M)$, and the standard deviation is

$$\sigma_{\hat{c}} = 2\sqrt{\frac{p(1-p)}{M}}.$$

The standard deviation attains its maximum at $p = 1/2$. In that case, $p(1-p) = 1/4$, and hence $\sigma_{\hat{c}} \leq 1/\sqrt{M}$. To achieve an additive error $\epsilon$ in estimating $l$, we thus require $1/\sqrt{M} \leq \epsilon$, which implies

$$M = \mathcal{O}\left(\frac{1}{\epsilon^2}\right).$$

## B.2 Determining Number of Qubits for the RBF Kernel

In this section, we derive the relationship between the number of qubits required, given by $\lceil \log_2(2D) \rceil$, and the additive error $\epsilon$ in approximating the RBF kernel. We consider the quantum feature map corresponding to the RBF kernel defined in (4).

$$x \in \mathbb{R}^d \mapsto \psi(x) = \frac{1}{\sqrt{D}} \sum_{j=1}^{D} \left( \cos\left(w_j^T x\right) |2j-2\rangle + \sin\left(w_j^T x\right) |2j-1\rangle \right).$$

Here, $w_j$ are drawn independently from the multivariate normal distribution $\mathcal{N}(0, \sigma^{-2}I)$. The associated kernel approximation is then given by

$$\hat{K}(x,y) = \langle \psi(x), \psi(y) \rangle = \frac{1}{D} \sum_{j=1}^{D} \cos\left(w_j^T (x-y)\right).$$

As shown in (7), the expectation of this approximation is

$$\mathbb{E}\left[\hat{K}(x,y)\right] = \exp\left(-\frac{1}{2\sigma^2} \|x-y\|^2\right) = K_{\text{RBF}}(x,y).$$

Define for each $j = 1, \ldots, D$, the random variables $X_j = \cos\left(w_j^T (x-y)\right)$. Then, the kernel approximation can be written as $\hat{K}(x,y) = \sum_{j=1}^{D} X_j / D$. Next, we compute the variance of each $X_j$. Let $Z_j = w_j^T (x-y)$. Because $w_j \sim \mathcal{N}(0, \sigma^{-2}I)$, it follows that $Z_j \sim \mathcal{N}\left(0, \|x-y\|^2/\sigma^2\right)$. Standard results for the cosine of a Gaussian random variable yield

$$\mathbb{E}[\cos(Z_j)] = \exp\left(-\frac{\|x-y\|^2}{2\sigma^2}\right), \quad \mathbb{E}[\cos^2(Z_j)] = \frac{1}{2}\left(1 + \exp\left(-\frac{\|x-y\|^2}{\sigma^2}\right)\right).$$

Thus, the variance of $X_j$ is

$$v := \text{Var}(X_j) = \mathbb{E}[\cos^2(Z_j)] - (\mathbb{E}[\cos(Z_j)])^2 = \frac{1}{2}\left[1 - \exp\left(-\frac{\|x-y\|^2}{\sigma^2}\right)\right].$$

Since the $X_j$ are i.i.d., the variance of the average $\hat{K}(x,y)$ is $\text{Var}\left(\hat{K}(x,y)\right) = v/D$.

We then define the centered random variables $Y_j := X_j - \mathbb{E}[X_j]$. Each $Y_j$ then has zero mean and variance $v$. Since $\cos(\cdot)$ is bounded in $[-1,1]$, it follows that $|X_j| \leq 1$ and $|X_j - \mathbb{E}[X_j]| = |Y_j| \leq 2$,. We then apply Bernstein's inequality (Bernstein, 1924) to the sum $\sum_{j=1}^{D} Y_j$, whose summands are bounded by $M$ (here $M = 2$), and whose total variance is $\sigma^2 = vD$. We have for any $t > 0$,

$$\Pr\left(\left|\sum_{j=1}^{D} Y_j\right| \geq t\right) \leq 2\exp\left(-\frac{t^2}{2vD + \frac{2}{3}Mt}\right).$$

Setting $t = D\epsilon$, we see

$$\Pr\left(\left|\hat{K}(x,y) - \mathbb{E}[\hat{K}(x,y)]\right| \geq \epsilon\right) \leq 2\exp\left(-\frac{D\epsilon^2}{2v + \frac{4}{3}\epsilon}\right).$$

For sufficiently small $\epsilon$ (i.e. when the term $\frac{4}{3}\epsilon$ is negligible relative to $2v$), this simplifies to

$$\Pr\left(\left|\hat{K}(x,y) - K_{\text{RBF}}(x,y)\right| \geq \epsilon\right) \leq 2\exp\left(-\frac{D\epsilon^2}{2v}\right).$$

To ensure that the additive error is bounded by $\epsilon$ with probability at least $1 - \delta$, we require

$$2 \exp\left(-\frac{D\epsilon^2}{2v}\right) \leq \delta.$$

Taking natural logarithms and rearranging, we obtain

$$D \geq \frac{2v}{\epsilon^2} \ln\left(\frac{2}{\delta}\right).$$

Thus, to guarantee an additive error of at most $\epsilon$ with confidence $1 - \delta$, the number of random features must satisfy

$$D = \mathcal{O}\left(\frac{v}{\epsilon^2}\right).$$

For large $\|x - y\|^2/\sigma^2$, we have $v = 1/2$, and for small $\|x - y\|^2/\sigma^2$, we can approximate the exponential term in $v$ with the first order of the Taylor expansion. Hence

$$D = \begin{cases} \mathcal{O}\left(\frac{1}{\epsilon^2}\right), & \text{for large } \|x-y\|^2/\sigma^2, \\ \mathcal{O}\left(\frac{\|x-y\|^2}{\sigma^2\epsilon^2}\right), & \text{for small } \|x-y\|^2/\sigma^2. \end{cases} \tag{12}$$

It is important to note that the constant $v$ depends on the ratio $\|x-y\|^2/\sigma^2$. For a fixed pair $(x, y)$, increasing $\sigma$ results in a smaller $v$. Consequently, for larger $\sigma$ (i.e., for a wider kernel), a smaller number of random features $D$ is required to achieve a given error tolerance $\epsilon$. Conversely, for smaller $\sigma$ (i.e., for a narrower kernel), the ratio $\|x - y\|^2/\sigma^2$ increases, leading to a larger $v$ and thus a larger $D$ is necessary. This can be seen in Figure 5(a).

Moreover, when $\sigma$ becomes very small (for example, when $\sigma < 0.5$) or when $\|x - y\|^2$ is very large (i.e., $x, y \in \mathbb{R}^d$, and $d$ is very large), the exponential term in $v$ rapidly approaches zero, and $v$ asymptotically approaches its maximum value of $1/2$. In this regime, the relationship between $D$ and $\epsilon$ becomes independent of $\sigma$. Consequently, the error curves will converge as we observe in Figure 5(c).

### B.3 Determining Number of Qubits for the Laplacian Kernel

In this section, we derive the relationship between the number of qubits required, given by $\lceil \log_2(2D) \rceil$, and the additive error $\epsilon$ in approximating the Laplacian kernel. We consider the quantum feature map corresponding to the Laplacian kernel defined in (8) as

$$x \in \mathbb{R}^d \mapsto \psi(x) = \frac{1}{\sqrt{D}} \sum_{j=1}^{D} \left(\cos\left(w_j^T x + \alpha_j\right) |2j - 2\rangle + \sin\left(w_j^T x + \alpha_j\right) |2j - 1\rangle\right),$$

where $w_j$ are drawn independently from the Cauchy distribution $\mathcal{C}(0, \alpha^{-1}I)$ and the phase shifts $\alpha_j$ are independent samples from the uniform distribution $\mathcal{U}(0, 2\pi)$. The associated kernel approximation is then given by

$$\hat{K}(x, y) = \langle \psi(x), \psi(y) \rangle = \frac{1}{D} \sum_{j=1}^{D} \cos\left(w_j^T(x - y)\right).$$

As shown in (10), the expectation of this approximation is

$$\mathbb{E}\left[\hat{K}(x, y)\right] = \exp\left(-\frac{\|x - y\|_1}{\alpha}\right) = K_L(x, y).$$

As before, define for each $j = 1, \ldots, D$, $X_j = \cos\left(w_j^T(x - y)\right)$. Then, the kernel approximation can be written as $\hat{K}(x, y) = \sum_{j=1}^{D} X_j/D$. Next, we compute the variance of each $X_j$. Let $Z_j = w_j^T(x - y)$. Since $w_j$ is drawn from $\mathcal{C}(0, \alpha^{-1}I)$ and Cauchy distributions remain closed under linear transformations, we have

$$Z_j \sim \mathcal{C}(0, \gamma), \quad \gamma = \frac{\|x - y\|_1}{\alpha}.$$

The characteristic function of a Cauchy random variable $Z_j \sim \mathcal{C}(0, \gamma)$ is given by $\phi_{Z_j}(t) = \exp\left(-\gamma |t|\right)$. Thus, setting $t = 1$ we obtain

$$\mathbb{E}[\cos(Z_j)] = \exp(-\gamma) = \exp\left(-\frac{\|x - y\|_1}{\alpha}\right).$$

Moreover, using the trigonometric identity $\cos^2(Z_j) = (1 + \cos(2Z_j))/2$, we have

$$\mathbb{E}[\cos^2(Z_j)] = \frac{1 + \exp(-2\gamma)}{2} = \frac{1 + \exp\left(-\frac{2\|x - y\|_1}{\alpha}\right)}{2}.$$

Thus, the variance of $X_j$ is

$$v := \mathrm{Var}(X_j) = \mathbb{E}[\cos^2(Z_j)] - (\mathbb{E}[\cos(Z_j)])^2 = \frac{1 - \exp\left(-\frac{2\|x - y\|_1}{\alpha}\right)}{2}.$$

Now, following the same procedure as in B.2 by defining $Y_j = X_j - \mathbb{E}[X_j]$ and applying Bernstein's inequality on $\sum_{j=1}^{D} Y_j$, we find that

$$D = \begin{cases} \mathcal{O}\left(\frac{1}{\epsilon^2}\right), & \text{for large } \|x - y\|_1/\alpha, \\ \mathcal{O}\left(\frac{\|x - y\|_1}{\alpha \epsilon^2}\right), & \text{for small } \|x - y\|_1/\alpha. \end{cases}$$

It is important to note that the constant $v$ depends on the ratio $\|x - y\|_1/\alpha$. For a fixed pair $(x, y)$, increasing $\alpha$ results in a small $v$. Consequently, for larger $\alpha$ (i.e., for a wider Laplacian kernel), the required $D$ to achieve a given error tolerance $\epsilon$ is smaller. Conversely, for a smaller $\alpha$ (i.e., for a narrower kernel), $v$ increases. Thus a larger $D$ is necessary. This can be seen in Figure 5(b). Similar to before, when $\alpha$ becomes very small (for example, when $\alpha < 1$) or when $\|x - y\|_1$ is very large (i.e., $x, y \in \mathbb{R}^d$, and $d$ is very large), the exponential term steadily approaches zero, and $v$ asymptotically approaches its maximum value of $1/2$. In this regime, the relationship between $D$ and $\epsilon$ becomes independent of $\alpha$ as we observe in Figure 5(d).

### B.4 Classical Simulation of Kernel Approximation Error

In this section, we describe experiments run on a classical computer to validate the theoretical claims made in Sections B.2 and B.3. Our goal is to assess the quality of the kernel approximations obtained via our feature maps for both the RBF and Laplacian kernels. To quantify the approximation quality, we compute the relative Frobenius norm error between the estimated kernel matrix and the exact kernel matrix. The error metric is defined as

$$\frac{\|K_{\text{exact}} - K_{\text{approx}}\|_F}{\|K_{\text{exact}}\|_F},$$

where $\|\cdot\|_F$ is the Frobenius norm.

Our empirical results as seen in Figure 5 confirm that the kernel approximation error decreases as the number of random features $D$ increases. In both the RBF and Laplacian kernel experiments, the observed decay rate aligns with the theoretical $\mathcal{O}(1/\sqrt{D})$ behavior predicted earlier. These findings, albeit using classical resources provide strong evidence that the proposed feature maps produce reliable approximations to exact kernel matrices.

## C  Broader Impact Statement and Ethical Concerns

Our work focuses on computing commonly used kernels in machine learning using quantum computing. As a piece of fundamental research, our contribution is theoretical in nature and does not, in itself, pose any direct negative societal impacts or ethical concerns. The methods and datasets involved do not contain sensitive personal data, nor do they target or adversely affect any vulnerable populations.

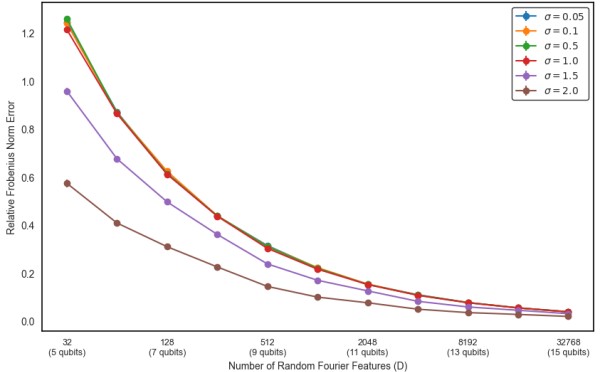

(a) RBF Kernel, $d = 10$. Different curves correspond to various values of $\sigma$, with the x-axis showing the number of random features $D$ (and corresponding qubits, $\lceil \log_2(2D) \rceil$).

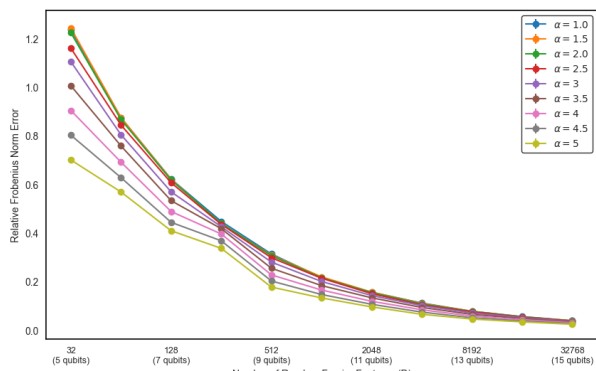

(b) Laplacian Kernel, $d = 10$. Different curves correspond to various values of $\alpha$, with the x-axis showing the number of random features $D$ (and corresponding qubits, $\lceil \log_2(2D) \rceil$).

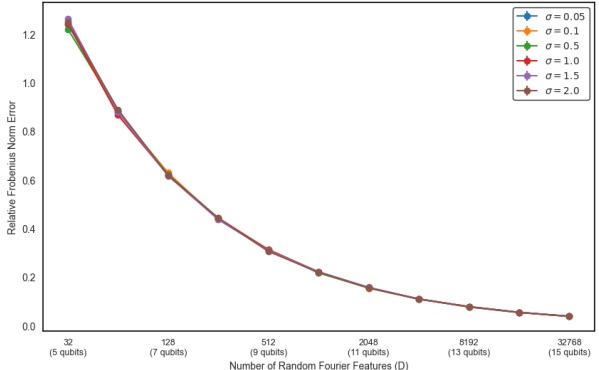

(c) RBF Kernel, $d = 100{,}000$. In this regime the $\|x - y\|^2$ term dominates, causing the variance to approach $1/2$ and all curves to coincide.

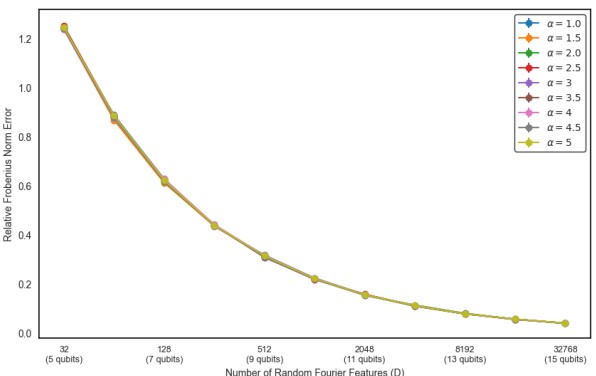

(d) Laplacian Kernel, $d = 100{,}000$. Again, the $\|x - y\|_1$ term dominates, causing the variance to approach $1/2$ and all curves to coincide.

Figure 5: Kernel approximation error curves for the RBF and Laplacian kernels under two data regimes. All experiments were performed on simulated datasets with 100 samples, and each experiment was repeated 5 times with the errors reported as averages. **Top:** Experiments with a small feature dimension ($d = 8$) show clear dependence on $\sigma$ (RBF) or $\alpha$ (Laplacian). **Bottom:** Experiments with a large feature dimension ($d = 100{,}000$) exhibit nearly overlapping curves regardless of kernel parameters, as the large $\|x - y\|$ values force the variance to saturate at $1/2$, resulting in $D = \mathcal{O}(1/\epsilon^2)$.

While it is acknowledged that kernel-based machine learning methods can, in certain applications, be associated with issues such as bias amplification or limited interpretability, especially when applied to non-curated datasets or in safety-critical systems, our work builds upon established techniques without introducing fundamentally new methods that would exacerbate these concerns. We recognize that downstream applications of kernel methods may encounter ethical challenges. However, such considerations are intrinsic to the broader field rather than a consequence of our specific contribution.

Given the theoretical scope of this work, no further mitigation strategies are necessary. Nevertheless, we urge practitioners to consider the ethical implications in any concrete application of these methods.

