# OpenReview forum: "Distributed and Secure Kernel-Based Quantum Machine Learning"
_TMLR — Accepted by TMLR_

### Review · Reviewer_ThpE · 2024-12-28

**Summary Of Contributions:**

This paper introduces a novel method for secure and distributed quantum kernel-based machine learning.  The authors introduce new quantum feature maps for polynomial, radial basis function (RBF), and Laplacian kernels, enabling secure computation of these kernels when data is distributed across multiple parties.  The proposed framework leverages quantum teleportation to ensure data security during transmission and is validated using a simulator from IBM (Qiskit Aer Simulator) on multiple public datasets.

**Audience:**

Yes

**Broader Impact Concerns:**

N/A.

**Claims And Evidence:**

Yes

**Requested Changes:**

Critical changes:
* Expand Kernel Evaluation: Implement and evaluate the proposed quantum feature maps for polynomial, RBF, and Laplacian kernels on the Qiskit Aer Simulator.  This would involve optimizing the quantum circuits for these kernels to fit within the simulator's qubit limitations or exploring alternative simulation platforms with higher qubit capacities.
* Explore Multi-Party Configurations: Extend the evaluation beyond the two-party configuration to include scenarios with three or more parties.

Proposed change for strengthening the work:
* Consider Malicious Entities: Extend the security analysis to consider actively malicious participants who may deviate from the protocol or attempt to compromise data privacy.  This could involve employing techniques from quantum cryptography and secure multi-party computation to enhance the framework's robustness against malicious attacks.

**Strengths And Weaknesses:**

Strengths:
* Novelty: The paper addresses a gap in the literature by extending quantum kernel-based machine learning to a distributed and secure setting.  The introduction of new quantum feature maps for commonly used kernels is a great contribution.
* Validation: The authors validate their approach through theoretical analysis and empirical evaluation using Qiskit Aer Simulator on publicly available datasets.

Weaknesses:
* Limited Evaluation: Due to limitations in the simulator, the evaluation focuses on linear kernels and a two-party configuration.  Testing other kernels and more complex configurations would provide a more comprehensive assessment of the approach.
* Semi-honest Adversary Model: The security analysis assumes semi-honest participants.  Exploring the framework's robustness against actively malicious entities would be beneficial.
* Lack of Clarity on Scalability: The paper does not explicitly discuss the scalability of the proposed approach for large datasets and complex machine learning tasks.  Providing insights into the computational and communication complexity as the number of parties and data dimensions increase would be valuable.
* Limited Discussion of Quantum Advantage: The paper briefly mentions the potential efficiency gains of quantum machine learning but does not provide a detailed analysis of the quantum advantage offered by the proposed approach.  A more in-depth discussion of the conditions under which the quantum method outperforms classical counterparts would strengthen the paper's contributions.
* Reliance on a Trusted Third Party: The architecture relies on a trusted third party (helper) to ensure the integrity of the Bell pairs used in quantum teleportation.  Exploring alternatives that reduce or eliminate the need for a trusted third party would enhance the framework's practicality and security.

---

> ### Author Response · Authors · 2025-02-12
> **Authors' Response to Reviewer ThpE (I)**
>
> We sincerely appreciate the reviewer’s thoughtful feedback and insightful comments. We are delighted that you find the introduction of relevant quantum feature maps a great contribution. Below, we provide point-by-point responses to address the concerns raised.
>
> > Limited Evaluation: Due to limitations in the simulator, the evaluation focuses on linear kernels and a two-party configuration. Testing other kernels and more complex configurations would provide a more comprehensive assessment of the approach.
>
> > Critical changes:
> >> Expand Kernel Evaluation: Implement and evaluate the proposed quantum feature maps for polynomial, RBF, and Laplacian kernels on the Qiskit Aer Simulator. This would involve optimizing the quantum circuits for these kernels to fit within the simulator's qubit limitations or exploring alternative simulation platforms with higher qubit capacities.
> >> Explore Multi-Party Configurations: Extend the evaluation beyond the two-party configuration to include scenarios with three or more parties.
>
> We acknowledge that our experimental evaluation is constrained by current quantum hardware limitations. Specifically, the use of the 31-qubit IBM Qiskit Aer Simulator necessitated a two-party configuration, with each participant representing their data with up to seven qubits, thereby restricting our evaluation primarily to linear kernels. However, our primary contribution lies in the introduction of quantum feature maps for some commonly used kernels and the formulation of a secure, distributed quantum kernel learning architecture.
>
> In response to your suggestions, **we have now provided code** to implement the polynomial, RBF, and Laplacian kernels, which will facilitate future evaluations as quantum hardware accessibility improves. Furthermore, our protocol is designed to be scalable beyond the two-party scenario; the _inner_product_and_teleport_ function is independent of the number of participants, making it readily extendable to multi-party settings.
>
> We have also made significant changes to our theoretical analysis (see **Chapter 5 and Appendix B**), where we discuss the computational complexity associated with our method. Here, we provide an idea of the number of qubits necessary to estimate the kernels using our quantum feature maps (B2, B3) and detail the number of shots that will be required to accurately compute the inner products (B1). Further, we also provided evidence for claims in Appendices B2 and B3 using classical resources in Appendix B4.
>
> > Semi-honest Adversary Model: The security analysis assumes semi-honest participants. Exploring the framework's robustness against actively malicious entities would be beneficial.
>
> > Proposed change for strengthening the work:
> >> Consider Malicious Entities: Extend the security analysis to consider actively malicious participants who may deviate from the protocol or attempt to compromise data privacy. This could involve employing techniques from quantum cryptography and secure multi-party computation to enhance the framework's robustness against malicious attacks.
>
> Our security analysis assumes a semi-honest adversarial model, which is a widely accepted standard in distributed and federated learning architectures [1,2]. In our work, it can be noted that malicious participants can still only alter the global model but cannot access the private data of other participants, as long as the server is non-colluding. Hence, our setup still maintains data privacy with malicious adversaries active. We have added a discussion regarding the same in **Section 6.3**.  However, we acknowledge that an adversarial colluding server could reveal the seed used by participants, enabling it to reconstruct the original data. We plan to explore such scenarios in future work.
>
>
> [1] Kairouz, Peter, et al. "Advances and open problems in federated learning." Foundations and trends® in machine learning 14.1–2 (2021): 1-210.
>
> [2] Goldreich, Oded. Foundations of cryptography: volume 2, basic applications. Vol. 2. Cambridge university press, 2001.

---

> ### Author Response · Authors · 2025-02-12
> **Authors' Response to Reviewer ThpE (II)**
>
> > Lack of Clarity on Scalability: The paper does not explicitly discuss the scalability of the proposed approach for large datasets and complex machine learning tasks. Providing insights into the computational and communication complexity as the number of parties and data dimensions increase would be valuable.
>
> > Limited Discussion of Quantum Advantage: The paper briefly mentions the potential efficiency gains of quantum machine learning but does not provide a detailed analysis of the quantum advantage offered by the proposed approach. A more in-depth discussion of the conditions under which the quantum method outperforms classical counterparts would strengthen the paper's contributions.
>
> Thank you for this suggestion. As mentioned above, we have now included **Section 5: Computational Complexity**, where we provide a theoretical analysis of the computational complexity of our quantum approach in comparison to its classical counterpart. While current hardware limitations prevent empirical validation using quantum hardware, we emphasize that our theoretical derivations remain sound. Further, we have also updated the work with **Appendix B** where we provide an in-depth theoretical analysis of the computational complexity associated with the number of shots and the number of qubits required in our protocol to illustrate what can be expected when working with quantum hardware to estimate the various kernels.
>
> > Reliance on a Trusted Third Party: The architecture relies on a trusted third party (helper) to ensure the integrity of the Bell pairs used in quantum teleportation. Exploring alternatives that reduce or eliminate the need for a trusted third party would enhance the framework's practicality and security.
>
> The role of the helper in our framework is strictly limited to preparing Bell states for quantum teleportation. If a trusted third party is unavailable, the server can assume this responsibility. The use of a trusted entity in security protocols however is a common practice in both quantum [3] and classical [4] secure computation frameworks. Exploring alternative approaches that eliminate the need for a trusted third party is an interesting avenue for future work.
>
> We appreciate the reviewer’s constructive feedback and believe that the additional clarifications and updates have strengthened our work. Thank you for your time and consideration.
>
> [3] Bennett, Charles H., et al. "Teleporting an unknown quantum state via dual classical and Einstein-Podolsky-Rosen channels." Physical review letters 70.13 (1993): 1895.
>
> [4] Canetti, Ran. "Security and composition of multiparty cryptographic protocols." Journal of CRYPTOLOGY 13 (2000): 143-202.

---

### Review · Reviewer_ec1v · 2025-01-06

**Summary Of Contributions:**

This work introduces quantum extensions to multiple standard kernel based machine learning methods. In doing so, the authors provide a way to perform machine learning on quantum computers in situations where the data is distributed using quantum resources such as quantum teleportation. These results are validated with a popular quantum simulation software, IBM Qiskit.

**Audience:**

Yes

**Claims And Evidence:**

No

**Requested Changes:**

1. Could you provide citations for the relevant algorithms in the introduction. For example, references to the quantum knn algorithm [1] or basic linear algebra [2] would make the work easier to follow for members of the community not familiar with quantum machine learning.
2. The introduction is imprecise in spots. For example, the introduction states that quantum computers yield representational benefits due to quantum superpositions, but it's been shown that multipartite entanglement is the only quantum resource necessary to achieve an exponential speedup over classical algorithms [3]. It would be nice to see the introduction cleaned up and made more precise.
3. There are multiple runon sentences in the introduction. `counterparts such as deep-learning, many kernel-based ML techniques often offer greater interpretability (Morocho-Cayamcela et al., 2019; Ponte & Melko, 2017), and provide better accuracy and models when the high-dimensional data is limited which is often the case in many real-world applications (Ding et al., 2021; Montesinos-López et al., 2021).` for example
4. The work could stand to be better motivated through a clearer invitation in the introduction
5. The last paragraph of the introduction is unclear. In particular, the two sentences that follow `although` are difficult to understand how they fit with the rest of the work.
6. Please expand the definition in eq1 to consider more than just $l_2$ regularization.
7. Please define and clarify what a computational basis is (bottom of page 3).
8. At the top of page 4, please clarify what `Copies of Quantum States` is? It's hard to follow.
9. Please clarify whether the architecture in 5.1 is a contribution of this work. if it is not, please cite it
10. Please clarify in 5.2.1 what a bell state is, and why they have to be bell states. Do you see performance drops when you have less than maximal entanglement?


[1] https://arxiv.org/abs/2003.09187v1
[2] https://link.springer.com/article/10.1007/s42484-021-00048-8
[3] https://arxiv.org/abs/quant-ph/0201143

**Strengths And Weaknesses:**

**Strengths**
1. The work is a novel contribution on an active area of research
2. The experiments are theoretically quite interesting

**Weaknesses**
1. The work is poorly motivated and poorly cited, making it hard to place the contribution within the field. Additionally, for a non-expert in the field, it can be challenging in spots to follow the work.
2. The work is in rough shape in spots, and I would recommend that the authors give the work an additional proof read. I've highlighted a non-exhaustive set of concerns in the requested changes section.
3. The related work section is quite brief. I would recommend that the authors highlight some of the seminal quantum ML papers, as well as quantum linear algebra papers, and some of the seminal papers for kernel methods. I'd also recommend extending the literature review to include other novel computational paradigms such as analog computers [1]. This isn't exhaustive, but I would invite the authors to perform a more thorough review of the literature.
4. The mathematics is hard to follow in spots. For example, the choice of computational basis states in the kernels (eg why $| 2 j -2 \rangle$ vs $| 2 j -1 \rangle$)

[1] https://www.nature.com/articles/s42256-024-00943-2

**Questions**

1. Are there practical considerations associated with Thm 1? How much precision would it require as a function of feature complexity?
2. In Thm 2, what assumptions are required to make the assumption that $w^t (x - y)$ is normally distributed?

---

> ### Author Response · Authors · 2025-02-12
> **Authors' Response to Reviewer ec1v (I)**
>
> We appreciate the reviewer’s thoughtful feedback and constructive suggestions. We are delighted that the reviewer finds the theoretical work interesting. Below, we provide point-by-point responses to address the concerns raised.
>
> **Comments regarding the Introduction:**
>
> > The work is poorly motivated and poorly cited, making it hard to place the contribution within the field. Additionally, for a non-expert in the field, it can be challenging in spots to follow the work.
>
> > The related work section is quite brief. I would recommend that the authors highlight some of the seminal quantum ML papers, as well as quantum linear algebra papers, and some of the seminal papers for kernel methods. I'd also recommend extending the literature review to include other novel computational paradigms such as analog computers. This isn't exhaustive, but I would invite the authors to perform a more thorough review of the literature.
>
> > The introduction is imprecise in spots. For example, the introduction states that quantum computers yield representational benefits due to quantum superpositions, but it's been shown that multipartite entanglement is the only quantum resource necessary to achieve an exponential speedup over classical algorithms. It would be nice to see the introduction cleaned up and made more precise.
>
> > There are multiple runon sentences in the introduction.
>
> > Could you provide citations for the relevant algorithms in the introduction. For example, references to the quantum knn algorithm or basic linear algebra would make the work easier to follow for members of the community not familiar with quantum machine learning.
>
> > The last paragraph of the introduction is unclear. In particular, the two sentences that follow although are difficult to understand how they fit with the rest of the work.
>
> > The work could stand to be better motivated through a clearer invitation in the introduction
>
> We have rewritten the Introduction and the Related Works to more clearly convey the motivation behind our study and to accurately place our contributions within the broader literature. In the revised Introduction, we explicitly discuss the role of entanglement in achieving speedups in quantum computing. Additionally, we have refined the text to eliminate run-on sentences and ambiguous phrasing. We have also added relevant citations to widen the literature review in the study.
>
> While we recognize the interest in alternative computational paradigms such as analog computing, we consider a discussion of these approaches to be beyond the scope of our study, which is focused on leveraging quantum computing to apply traditional machine learning techniques.
>
> > The mathematics is hard to follow in spots. For example, the choice of computational basis states in the kernels (eg why $|2j-2>$ vs $|2j-1>$)
>
> Our choice of basis states is designed to capture both the cosine and sine components required for our quantum feature map. Specifically, as $j$ ranges from $1$ to $D$, the states $|2j-2>$ and $|2j-1>$ collectively cover the interval from $0$ to $2D-1$.
>
> **Responses to Questions**
>
> > Are there practical considerations associated with Thm 1? How much precision would it require as a function of feature complexity?
>
> In the computation of the polynomial kernel,  the precision is related to the number of measurement shots required to estimate the kernels. This is determined by the precision of the Fredkin gate employed in our protocol. Specifically, to achieve an error of $\epsilon$, the number of shots scales as $\mathcal{O}(1/\epsilon^2)$. We have expanded this discussion in Appendix B1 for further clarity. Related to this, the precision in computing the RBF and Laplacian kernels depends not only on the number of shots of the circuit but also on the number of random Fourier features (hence qubits) used, which is now detailed in Appendices B2 and B3.
>
> > In Thm 2, what assumptions are required to make the assumption that $w^T(x−y)$ is normally distributed?
>
> The normality of $w_j^T(x-y)$ is not an ad hoc assumption but follows directly from our quantum feature map definition, where each component $w_j$ is independently sampled from a normal distribution. We have reworded the relevant section to clarify this and remove any ambiguity.

---

> ### Author Response · Authors · 2025-02-12
> **Authors' Response to Reviewer ec1v (II)**
>
> **Requested Changes and Updates**
>
> > Please expand the definition in eq1 to consider more than just $l_2$ regularization.
>
> The definition now incorporates other forms of regularization. Thank you for pointing this out.
>
> > Please define and clarify what a computational basis is.
>
> Added a definition in Section 2.3 now.
>
> > At the top of page 4, please clarify what Copies of Quantum States is? It's hard to follow.
>
> This part of Section 3.1 is now rewritten for readability.
>
> > Please clarify whether the architecture in 5.1 is a contribution of this work. if it is not, please cite it
>
> We have now explicitly stated that our architecture builds upon existing quantum protocols such as quantum teleportation and Fredkin gates. This has been clarified both in the introduction and Section 6, where we formally define and extend prior work [1].
>
> > Please clarify in 5.2.1 what a bell state is, and why they have to be bell states. Do you see performance drops when you have less than maximal entanglement?
>
> Regarding the impact of reduced entanglement: Since our experiments are limited to simulators, we cannot measure the precise impact. However, our experiments on Qiskit’s Aer Simulator with depolarizing noise suggest that accuracy degrades with reduced entanglement, as expected. This aligns with the theoretical expectation that failed quantum teleportation disrupts data integrity.
>
> We appreciate the reviewer’s valuable feedback and believe that the revisions have significantly improved the clarity and rigor of our manuscript. Thank you for your time and consideration.
>
> [1] Sheng, Yu-Bo, and Lan Zhou. "Distributed secure quantum machine learning." Science Bulletin 62.14 (2017): 1025-1029.

---

### Review · Reviewer_vwmN · 2025-02-04

**Summary Of Contributions:**

This paper presents a method for performing kernel-based quantum machine learning on distributed datasets in a secure manner.  The authors introduce quantum feature maps for polynomial, radial basis function (RBF), and Laplacian kernels, proving their correctness theoretically.  They propose a secure architecture utilizing quantum teleportation for distributed kernel computation.  The architecture is validated for linear kernels on public datasets using Qiskit's Aer Simulator.

**Audience:**

Yes

**Broader Impact Concerns:**

No statement found about Broader Impact Statement in the paper.

**Claims And Evidence:**

No

**Requested Changes:**

Refer Weaknesses

**Strengths And Weaknesses:**

**Strengths**
1. The paper is easy to follow and understand, even for readers without a strong background in quantum computing.
2.  The paper addresses the important problem of secure and distributed quantum machine learning.
3. The proposed approach is potentially practical and could be implemented on real quantum devices.



**Weaknesses**
1. While the paper introduces quantum feature maps for polynomial, RBF, and Laplacian kernels, the underlying principle of encoding classical data into quantum states to implicitly define a kernel is not novel.  This work extends the concept to new kernels, but the novelty lies in the specific construction of each map, not a fundamental shift in how quantum feature maps are obtained.
For example, the polynomial kernel's feature map utilizes the multinomial theorem to ensure the resulting quantum state is properly normalized, which is a direct translation of the classical feature space into a quantum state space.  Similarly, the RBF and Laplacian kernel feature maps leverage Random Fourier Features (RFF) to approximate the kernel function in a quantum state.  These constructions are technically sound but primarily apply existing principles.
2. The building blocks for security in this paper are not new.  They use existing tools like quantum teleportation, which is a known method for transferring quantum information securely.  They also rely on a "helper" to set things up, which is a common approach in secure quantum systems.   While these tools are effective, the paper doesn't introduce new ways of achieving security.
3. The system is designed for a specific scenario where everyone mostly follows the rules.  It assumes that the different parties involved might try to peek at information they shouldn't have access to, but they won't try to completely break the system or cheat in a major way.
This is a limited type of security because it doesn't protect against more serious attacks where someone might intentionally try to disrupt the system or steal information.
4. The authors have mentioned "previous work did not identify or utilize the implicit relationship between quantum encoding and kernels". However, there are many works starting from Maria’s work that use squeeze states or other nonlinear coherent states to define quantum kernel methods, to several other works that investigated by proposing new quantum kernels. For example, recent work on Non-Linear Optical Reproducing Kernels, operates by nonlinearly transforming data into feature space constructed with quantum states. They present a novel feature space constructed using Kerr coherent states, which generalize su(2), su(1, 1) coherent states, and squeezed states.
5. he empirical validation is restricted to linear kernels due to limitations in simulating large-scale quantum systems.  This limitation raises concerns about the architecture's practicality for more complex kernels with higher qubit requirements.
6. The experiments involve only two parties due to qubit constraints.  While the approach is theoretically extendable, the lack of empirical evidence for multi-party computation raises questions about scalability and performance in larger, more complex distributed settings.

---

> ### Author Response · Authors · 2025-02-12
> **Authors' Response to Reviewer vwmN (I)**
>
> We sincerely appreciate the reviewer’s thoughtful feedback and the acknowledgment of our work’s strengths. We are delighted that the paper was easy to read and follow. Below, we address the concerns raised in detail.
>
> > While the paper introduces quantum feature maps for polynomial, RBF, and Laplacian kernels, the underlying principle of encoding classical data into quantum states to implicitly define a kernel is not novel. This work extends the concept to new kernels, but the novelty lies in the specific construction of each map, not a fundamental shift in how quantum feature maps are obtained. For example, the polynomial kernel's feature map utilizes the multinomial theorem to ensure the resulting quantum state is properly normalized, which is a direct translation of the classical feature space into a quantum state space.
>
> We fully acknowledge that encoding classical data into quantum states to implicitly define a kernel is not a novel concept and was first proposed in [1]. As originally noted in the Introduction and now further detailed in our **expanded Related Works section**, our contribution lies in extending this established principle by constructing specific quantum feature maps for the polynomial, RBF, and Laplacian kernels, just as observed by the reviewer.
>
> With regards to the polynomial kernel, while [1] introduced the quantum feature map corresponding to the homogeneous polynomial kernel, our literature review did not reveal any prior work that constructs the polynomial kernel itself using quantum encodings. This gap motivated us to include this particular case in our study.
>
> While we agree Random Fourier Features (RFFs) are a well-established method in classical machine learning for approximating kernel functions, their adaptation to the quantum domain is novel. This is especially interesting since they are used to reduce dimensionality in the classical setting, while we use them to determine the number of qubits used to estimate the kernel.
> In summary, our work provides valuable contributions by extending existing kernel computations into the quantum domain and integrating these approaches with a secure distributed learning architecture.
>
> > The building blocks for security in this paper are not new. They use existing tools like quantum teleportation, which is a known method for transferring quantum information securely. They also rely on a "helper" to set things up, which is a common approach in secure quantum systems. While these tools are effective, the paper doesn't introduce new ways of achieving security.
>
> We acknowledge that the building blocks of our architecture such as quantum teleportation and the helper-assisted setup, are not novel.
>
> However, our contribution lies in the formalization of a secure distributed setting extending the architecture from [2] to a more general setting. This is now explicitly discussed in both the introduction and Section 6 to articulate how our architecture integrates known methods to achieve secure multi-party kernel computation. Our aim was never to introduce a fundamentally new security mechanism but rather to demonstrate how existing techniques can be adapted and extended to a particular setting.
>
> > The system is designed for a specific scenario where everyone mostly follows the rules. It assumes that the different parties involved might try to peek at information they shouldn't have access to, but they won't try to completely break the system or cheat in a major way. This is a limited type of security because it doesn't protect against more serious attacks where someone might intentionally try to disrupt the system or steal information.
>
> Our security analysis assumes a semi-honest adversarial model, which is a widely accepted standard in distributed and federated learning architectures [3,4]. In our work, it can be noted that malicious participants can still only alter the global model but cannot access the private data of other participants, as long as the server is non-colluding. Hence, our setup still maintains data privacy with malicious adversaries active. We have added a discussion regarding the same in **Section 6.3**. However, we acknowledge that an adversarial colluding server could reveal the seed used by participants, enabling it to reconstruct the original data. We plan to explore such scenarios in future work.
>
> [1] Schuld, Maria, and Nathan Killoran. "Quantum machine learning in feature hilbert spaces." Physical review letters 122.4 (2019): 040504.
>
> [2] Sheng, Yu-Bo, and Lan Zhou. "Distributed secure quantum machine learning." Science Bulletin 62.14 (2017): 1025-1029.
>
> [3] Kairouz, Peter, et al. "Advances and open problems in federated learning." Foundations and trends® in machine learning 14.1–2 (2021): 1-210.
>
> [4] Goldreich, Oded. Foundations of cryptography: volume 2, basic applications. Vol. 2. Cambridge university press, 2001.

---

> ### Author Response · Authors · 2025-02-12
> **Authors' Response to Reviewer vwmN (II)**
>
> > The authors have mentioned "previous work did not identify or utilize the implicit relationship between quantum encoding and kernels".
> However, there are many works starting from Maria’s work that use squeeze states or other nonlinear coherent states to define quantum kernel methods, to several other works that investigated by proposing new quantum kernels.
>
> The reviewer’s concern arises from a misinterpretation of our statement regarding previous works. Our comment referred specifically to [2], which implemented distributed secure kernel learning to compute the linear kernel without explicitly acknowledging or leveraging the implicit connection between quantum feature encodings and kernel functions. To avoid misunderstandings, we have reworded this in the Introduction and the Related Works sections of the manuscript to avoid any potential ambiguity.
>
> > The empirical validation is restricted to linear kernels due to limitations in simulating large-scale quantum systems. This limitation raises concerns about the architecture's practicality for more complex kernels with higher qubit requirements.
>
> > The experiments involve only two parties due to qubit constraints. While the approach is theoretically extendable, the lack of empirical evidence for multi-party computation raises questions about scalability and performance in larger, more complex distributed settings.
>
> We acknowledge that our experimental evaluation is constrained by current quantum hardware limitations. Specifically, the use of the 31-qubit IBM Qiskit Aer Simulator necessitated a two-party configuration, with each participant representing their data with up to seven qubits, thereby restricting our evaluation primarily to linear kernels. However, our primary contribution lies in the introduction of quantum feature maps for some commonly used kernels and the formulation of a secure, distributed quantum kernel learning architecture.
>
> In response to your suggestions, **we have now provided code** to implement the polynomial, RBF, and Laplacian kernels, which will facilitate future evaluations as quantum hardware accessibility improves. Furthermore, our protocol is designed to be scalable beyond the two-party scenario; the _inner_product_and_teleport_ function is independent of the number of participants, making it readily extendable to multi-party settings.
>
> We have also made significant changes to our theoretical analysis (see **Chapter 5 and Appendix B**), where we discuss the computational complexity associated with our method. Here, we provide an idea of the number of qubits necessary to estimate the kernels using our quantum feature maps (B2, B3) and detail the number of shots that will be required to accurately compute the inner products (B1). Further, we also provided evidence for claims in Appendices B2 and B3 using classical resources in Appendix B4.
>
> > No statement found about Broader Impact Statement in the paper.
>
> A Broader Impact Statement has now been added to **Appendix C**.
>
> We appreciate the reviewer’s valuable feedback and believe that the revisions have significantly improved the clarity and rigor of our manuscript. Thank you for your time and consideration.
>
> [2] Sheng, Yu-Bo, and Lan Zhou. "Distributed secure quantum machine learning." Science Bulletin 62.14 (2017): 1025-1029.

---

### Author Response · Authors · 2025-02-12
**Revision Summary**

Dear Reviewers,

We would like to express our sincere gratitude for the thoughtful and constructive feedback provided during the review process. In response, we have carefully revised our manuscript to address the comments and enhance the overall quality of our work. Below is a summary of the major and minor changes we have made since the last submission:

Major Changes:

1. Expanded the discussion of previous works that motivated our study in the Related Works section.

2. We have incorporated an extensive theoretical analysis comparing the computational complexity of our quantum approach to that of its classical counterpart. This analysis is detailed in Chapter 5 and further discussed in Appendix B.

3. Extended appendices -

    3.1. Appendix B1: Number of shots required for the protocol.

    3.2. Appendix B2 & B3: Number of qubits required for the RBF and Laplacian kernels.

    3.3. Appendix B4: Classical experiments illustrating and validating claims about qubit requirements from B2 and B3.

    3.4. Appendix C: Broader Impact Statement

4. Code Availability: We have added complete code for the polynomial, RBF, and Laplacian kernels to our anonymized GitHub repository.

Minor Changes:

1. Rephrased any confusing terminology and improved explanations of contributions.

2. Fixed run-on sentences, and reworded/rephrased sentences throughout the text to enhance readability.

3. Added definitions for computational basis and copies of quantum states for better comprehension.

4. Briefly addressed malicious adversarial considerations and their impact on security and model robustness.

An updated manuscript is now available, with all revisions/changes highlighted in blue. We hope these changes effectively address your concerns and further strengthen our submission. Thank you once again for your valuable feedback and time.

We are available to answer any further questions.

Sincerely, TMLR Paper3549 Authors

---

### Decision · Action_Editor_M2kC · 2025-03-29

**Recommendation:** Accept as is

**Comment:**

The contribution seems minor -- connecting two domains, but worth writing down.  It is not clear if it will ultimately have applications, and the most interesting part was not able to be simulated.  But the authors provided some code to eventually enable it.

See more discussions above on the Claims of Evidence and Audience.

**Audience:**

Yes, this is a classic machine learning topic -- in the context of quantum.   People with no quantum background may not get much out of it.  But it seems relevant to quantum machine learning.

**Claims And Evidence:**

I believe the paper satisfies the claims and evidence issue.  There are still two concerns:  novelty of the kernel feature representation in quantum setting, and evaluation beyond linear kernels.
In my reading is that the literature review is now fairly and clear, there are at the least slightly more explicit detail in the description and proofs in this paper.
As far as experiments, I agree with the lingering concerns about how this will work beyond linear kernels -- this is where kernel methods get most of their benefit.  But this requires quite large dimensions, and the available and accessible qBits are far behind where is needed to make this useful.  But so was the case in classical setting 20-30 years ago.

Ultimately, I found the paper well written, and it appears there are - at least some - novel contributions, which are put in context, and described clearly.

---

> ### Author Response · Authors · 2025-04-01
> **Authors' Response to Action Editor M2kC**
>
> Dear Action Editor,
>
> We would like to thank you and the reviewers for the time and effort spent in reviewing our submission. We are grateful for the thoughtful feedback and for accepting the paper.
>
> We have now uploaded the camera-ready version of the paper.
>
> Thank you again for your support and for handling our submission.
>
> Sincerely, TMLR Paper3549 Authors